# Mass Spectrometry Imaging Spatial Tissue Analysis toward Personalized Medicine

**DOI:** 10.3390/life12071037

**Published:** 2022-07-12

**Authors:** Juliana P. L. Gonçalves, Christine Bollwein, Kristina Schwamborn

**Affiliations:** 1Institute of Pathology, School of Medicine, Technical University of Munich, Trogerstraße 18, 81675 Munich, Germany; christine.bollwein@tum.de; 2German Cancer Consortium (DKTK), Partner Site Munich, 80336 Munich, Germany

**Keywords:** mass spectrometry imaging, proteomics, pathology, personalized medicine, cancer research, tissue typing

## Abstract

Novel profiling methodologies are redefining the diagnostic capabilities and therapeutic approaches towards more precise and personalized healthcare. Complementary information can be obtained from different omic approaches in combination with the traditional macro- and microscopic analysis of the tissue, providing a more complete assessment of the disease. Mass spectrometry imaging, as a tissue typing approach, provides information on the molecular level directly measured from the tissue. Lipids, metabolites, glycans, and proteins can be used for better understanding imbalances in the DNA to RNA to protein translation, which leads to aberrant cellular behavior. Several studies have explored the capabilities of this technology to be applied to tumor subtyping, patient prognosis, and tissue profiling for intraoperative tissue evaluation. In the future, intercenter studies may provide the needed confirmation on the reproducibility, robustness, and applicability of the developed classification models for tissue characterization to assist in disease management.

## 1. Introduction

As more information about disease development and particularly neoplasias is gathered, it becomes more evident that there are individual differences that set the course for disease development and treatment outcome. These individual differences are continuously being reported, and as a result, new tumor entities are defined. As a consequence, the number of tumor entities described over the last few years has increased drastically, and alongside it, the number of tests for its identification [1,2,3,4]. The successful identification of the tumor particularities at the time of diagnosis can set the course of the treatment and directly impact the patient’s quality of life. However, recognizing microscopic differences in some tumor entities is not straightforward, even for very experienced medical professionals. As a conservative approach, unknown subentities or inconclusive diagnoses are often inadequately treated, carrying a heavy burden on the patient’s mental and physical health.

Human bodily functions are dictated by the balanced translation of DNA to RNA to proteins. During that translation, errors can occur, which result in amino acid polymorphisms, protein isoforms, and post-translational modifications (PTMs). These proteoforms have distinct molecular activity and can modulate different cellular functions, driving the cell processes and signals to malfunction. This can result in, for example, the misfolding of proteins, which is associated with various diseases such as cancer and Alzheimer’s disease [5,6,7]. Further insight into these modifications has led to major breakthroughs in comprehending disease evolution, namely, cancer growth and development [8,9,10]. Therefore, studying proteins in the context of cancer is required to further understand the biology and molecular subtleties of a certain cancer subtypes, and predict their development and response to therapy. In clinical pathology, the diagnosis is usually carried out with the use of a wide range of tests and technologies (e.g., immunohistochemistry staining) based on the proteomic activity of the resected tissue specimen. However, the current approaches can be ambiguous, time-consuming, financially expensive, and require a significant amount of tissue to obtain a complete diagnosis.

To analyze and quantify proteins, the favored methodology is mass spectrometry [11,12]. Due to its high precision, reproducibility, and robustness, this approach has been part of proteomics research laboratories for many decades. While obtained results from traditional mass spectrometry are very accurate and reproducible, to analyze tissue samples, a preliminary evaluation is required to identify the tissue’s regions or cell population of interest, followed by laborious sample preparation. As an alternative, mass spectrometry imaging (MSI) has been proposed.

## 2. Mass Spectrometry Imaging (MSI)

Less invasive approaches such as blood and urine collection would ideally be used for the assessment of diseases. However, when it comes to tumor analysis, it is important to directly evaluate the tissue, as protein and metabolite concentrations can largely vary at the different locations in the tissue, and such discrepancies are not always translated to the urine or the plasma. Furthermore, the analysis of the tumor microenvironment plays an important role in tumor development and particularly in local resistance, immune escaping, and the occurrence of distant metastasis [13,14]. In addition, tissues, and notably tumors, are spatially complex leading to intra- and intertumoral heterogeneity. Linking molecular information and spatial distribution with tissue morphology is essential for a correct and concise biological interpretation [15,16]. To this end, MSI can be an omnipotent approach, since it has been developed with the purpose of evaluating analytes directly from tissue sections without the need for a previous microscopic investigation of the tissue or complex sample preparation protocols, and to allow for the correlation of molecular signatures with tissue morphology (Figure 1).

It is common practice in the clinic that samples are fixated in formalin immediately after material resection even before reaching the pathology lab for evaluation. For this reason, MSI protocols have been adapted to be integrated into the general clinical workflow, keeping a simple yet reproducible sample preparation for formalin-fixed paraffin-embedded (FFPE) samples [17].

In mass spectrometry approaches, the instrumentation is composed of three main modules: the ionization source, the mass analyzer, and the ion detector [18]. What sets MSI apart from traditional mass spectrometry analysis is that the precise position of the ionized analytes is recorded, and a mass spectrum is generated (Figure 1). This facilitates the correlation between the histological features and the precise location of the analytes in the tissue. In the first data acquisition step, analytes are ionized and separated according to their mass to charge (*m*/*z*), from where the molecular weight can be derived before reaching a detector, where the resulting ions are counted. By the end of the run, the user obtains a mass-spectrum derived from the x,y position where the molecules were ionized (the laser acquired the spectrum) within the sample [18]. The same information can also be recorded as a matrix, where the intensity of the individual molecular ions and their position in the tissue are saved by the collected measurement region or pixel (Figure 2). The distribution of a specific *m*/*z* value in the tissue can be visualized by generating an intensity map of the intended region. In addition, it is possible to compare the intensity of a particular molecular ion in different regions of the tissue (Figure 2e). Subsequently, the molecules of interest can be identified by tandem mass spectrometry and data comparison with established databases or by de novo sequencing.

There is a growing number of MSI approaches, in particular with regards to ambient ionization techniques, utilizing different desorption and ionization sources that offer varying resolutions and analyte classes [19].

MSI has been achieving major developments in different fields with a wide array of applications that are not limited to medicine. For instance, it has been employed for the visualization of the distribution of food components and identifying food factors, leading to higher quality assurance, better food safety, and more accurate nutritional analysis [20]. In plant biology, MSI has been making great progress over the last decade, where diverse studies have been carried out to further understand the fundamental plant biology [21]. In the field of forensic sciences, MSI has been utilized for the direct analysis of fingerprints for the identification of dermal contact with drugs, cosmetics, or specific foods [22,23]. However, most efforts have been directed to medical research, namely, tumor analysis.

### 2.1. Mass Spectrometry Imaging for Proteomic Profiling of Cancer

Notwithstanding the breakthroughs in understanding cancer, diagnosis, and disease management over the last couple of decades, the disease evolution remains to a great extent a mystery [24,25,26]. The mechanisms of cancer development and disease progression are still not fully comprehended [27,28,29]. The number of identified cancer subtypes is ever increasing due to the discovery of new singularities such as a genetic mutation or a disease-specific biomarker. As a consequence, a more precise diagnostic and subsequent treatment course can be devised. Typically, to reach a precise diagnosis, a series of immunohistological tests are required, followed by a detailed evaluation by a pathologist, which is often very time-consuming, delaying the start of the time-sensitive treatment. On top of that, the process can require a significant amount of tissue for the analysis, which is often not available in tumor biopsy samples.

For proteomic profiling using mass spectrometry imaging, the amount of required tissue is very small, namely, a single 2–3 μm thick tissue section that can afterwards still be used for histological or immunohistochemical evaluation, DNA analysis, or further proteomic investigation [30,31,32,33]. Additionally, the number of measured analytes in a single run is considerably higher when compared to current methods where one or a small set of protein signatures is evaluated in one measurement. For these reasons, a significant part of efforts has been directed towards training machine-learning (ML) methods using the proteomic or metabolomic profiles obtained directly from the tissue by MSI. ML algorithms look for traits in the data that can categorize the information in subgroups or classes either by employing unsupervised machine learning where the data are grouped by their affinity without requiring any preliminary analysis of the sample or through the use of supervised ML algorithms where previously acquired information (e.g., tumor diagnosis, staging, or treatment outcome) is used to train the algorithms. In the unsupervised approach, the outcome is the division of the data into clusters, with the significance of the division being given by the analyst; in the supervised approach, the outcome is already given by the algorithm on the basis of input information. Both approaches have yielded very good results. For instance, unsupervised ML often performs very well in distinguishing normal tissue from tumor tissue. However, when it comes to disease prognosis and closely resembling tumor entities, supervised machine learning seems to generate more accurate predictions.

One of the clinical applications of MSI is to assist in intraoperative consultation for the assessment of tumor margins to guarantee complete tumor resection [34,35,36]. One of the hallmarks in tissue diagnosis is the development of small devices that can quickly differentiate tumor from normal tissue, such as the iKnife and MasSpec Pen. These devices guide the surgeons by quickly classifying the tissue while performing resection surgery of various organs (such as the colon, breast, stomach, liver, lung, and brain) [37,38,39,40]. The apparatus relies on mass spectrometry measurements associated with ML algorithms to discriminate the tissue material from frozen tissue sections or directly in vivo [41,42,43]. As an example, King et al. utilized ambient ionization mass spectrometry to ionize samples ex vivo to assist in the differentiation of tumor resection margins of pancreatic ductal adenocarcinoma (PDAC) (*n* = 53) from the bile duct (*n* = 23) and nontumor samples of pancreatic tissue (*n* = 58). Samples collected prospectively were measured using the MasSpec Pen technology, and used for training and testing the least absolute shrinkage and selection operator (lasso) classifiers. The method yielded over 98% agreement with histological evaluation when classifying normal pancreas and PDAC in the training set (78 tissue analyses; PDAC samples with >70% tumor cells) and 78.8% in the validation set (33 tissue analyses; samples with mixed cellularity and low epithelial tumor cell concentration) [44]. A second lasso classifier was employed to differentiate bile duct (*n* = 16) and PDAC (*n* = 27) with accuracy of 98% in the training set, and accuracy of 91% in the validation set (bile duct, *n* = 7; PDCA, *n* = 17; PDAC invading bile duct, *n* = 8). Classifiers built on ex vivo samples were then utilized to classify tissue in the operating room in vivo and ex vivo (64 analyses), achieving an overall agreement of 93.8%. These studies consolidate that molecular signatures detected by MSI can be applied in real time to discern tumor margins and tissue with different provenience within approximately only 3 seconds [45].

The differentiation between malignant and benign skin lesions also presents a challenge that can be addressed by MSI characterization. Margulis et al. applied DESI–MSI to measure the lipid and metabolite profile of basal cell carcinoma (BCC), a common skin cancer, and normal skin, with the objective of identifying micrometer-sized tumor aggregates of malignant skin lesions [46]. Arachidonic acid and glycerophosphoglycerol were markedly abundant in BCC compared to normal skin regions [46]. In this study, a lasso regression based on solely 24 mass features was able to discriminate BCC aggregates from adjacent normal skin. The authors reported that this approach could be employed as a fast intraoperative process during Mohs surgeries, complementing the histopathological evaluation.

Due to the histological similarities between some tumor entities and the lack of specific markers, it can be challenging and very time-consuming for pathologists to reach a diagnosis. Medulloblastoma and pineoblastoma share clinical features and show identical histological characteristics [47]. The analysis of the lipid profile of pediatric medulloblastoma and pineoblastoma indicated that MALDI–MSI could be a suitable tool to support the diagnosis. To further understand both tumor types, the authors of this study performed receiver operating characteristic (ROC) analysis of the mass spectrometry features, concluding that glycerophosphoglycerols and glycerophosphocholines exhibited higher intensity, and could, therefore, become potential markers for medulloblastoma, while sphingolipids showed higher expression in pineoblastoma [47].

Another diagnostic conundrum in clinical pathology is to correctly characterize chromophobe renal cell carcinoma (chRCC) from renal oncocytoma (RO). RO is a benign kidney lesion that, from a histomorphological perspective, closely resembles the malignant neoplasia chRCC. This can result in the overtreatment of RO patients. On the basis of metabolite and lipid profiles obtained by the DESI–MSI of 71 patients with renal cell neoplasia, Zhang et al. were able to discriminate benign from malignant tumors with 100% accuracy [48].

PDAC also exhibits close morphology and histological resemblance to cholangiocarcinoma (CC). Both entities arising from the epithelium of the pancreaticobiliary tree have aggressive behavior, and an incorrect diagnosis can have strong implications on the patient’s prognosis and therapeutic course. Bollwein et al. utilized the proteomic profile of 82 patients measured by MALDI–TOF to train and test classification algorithms to differentiate between the two tumor types with accuracy of approximately 90% [49]. The authors also advanced a feature importance list calculated by the mean decrease in the impurity of gradient-boosting classification, which revealed that histone H2A and the collagen α-1 (I) chain are more intensely expressed in PDAC when compared with CC, which could be disease-related biomarkers [49].

MSI has matured and is taking confident strides in assisting in tumor diagnosis by either providing further understanding of the molecular composition of complex tumor structures or merely helping in simpler tasks such as the identification of tumor regions; studies applying MSI have presented encouraging results that it could be the clinicians’ right hand for tumor analysis and classification [31,50].

### 2.2. Prognosis Studies of Tumor Using Mass Spectrometry Imaging

An essential part of the clinical diagnosis of tumors is to predict how the tumor affects the patient’s outcome, and to predict the patient’s response to a certain treatment. For that, assessing the development stage of the disease is imperative, but it is also necessary to characterize the predictive molecular variation within the patient’s tumor to foresee the reactivity of a certain treatment.

MSI-based models were explored as a tool to accurately evaluate predictive molecular variations [51,52,53]. In a recent study by Erlmeier et al., MALDI–MSI was used in correlation with Kaplan–Meier curves to estimate predictive metabolic profiles for the prognosis of renal cell carcinoma (RCC) [54]. An increase in nucleotides (namely, cyclic guanosine monophosphate) was associated with a poor prognosis. The authors were also able to detect some metabolic pathways specific to some tumor types, particularly the glutathione metabolism, which is increased in late-stage clear cell RCC and associated with poor outcomes [54].

As the range of molecules characterized by MSI is not restricted to metabolites, prognostic studies based on glycan, protein, and lipid activity were carried out as well. One of these studies was performed by Phillips and coworkers, where prognostic features of triple-negative breast cancer (TNBC) were evaluated through the analysis of tryptic digested proteins utilizing a MALDI–TOF–TOF instrument [55]. The authors were able to identify 14 proteins that distinguish TNBC from benign lesions, and the correlation between these proteins and the Kaplan–Meier curves showed that COL1A1, COL1A2, COL6A3, ATIC, CCDC24, PLEKHG2, SOX11, and UBR4 are correlated with poor patient outcomes [55]. The results are supported by the literature, as COL1A1 and COL1A2, two components of Type I collagen, are upregulated in invasive breast cancer, with a potential role in spinal metastasis [56]. Aminoimidazole carboxamide ribonucleotide transformylase/inosine monophosphate cyclohydrolase (ATIC) is also necessary for cell proliferation [57], and SOX11 plays a role in breast cancer growth and invasion, and in regulating the basal-like phenotype [58].

The diagnosis and prognosis of prostate cancer are performed on the basis of histological evaluation following the Gleason scoring system. The Gleason score (GS) is based on the evaluation of changes in the morphology of tumor glands, but it does not provide any information about the metabolic pathways that caused the alteration. To explain the metabolic alterations, studies correlating the GS and molecular features have been carried out. The tryptic digestion of the tissue microarrays of 729 human prostate cancer specimens measured by MALDI–TOF–MSI identified four molecular signatures associated with a low Gleason score, early disease stage, and low proliferation marker Ki-67. One molecular feature was associated with high Ki-67, and another signal could be correlated with a prolonged time for prostate-specific antigen (PSA) recurrence [59]. In an independent pilot study, the lipid profile of prostate cancer samples was correlated with the Gleason score. The authors showed that phosphatidylcholines, phosphatidic acids, phosphatidylserines, phosphatidylinositols, and cardiolipins were overexpressed in GS (4 + 3), suggesting their involvement in the disease progression, and the possibility for them to be utilized as markers of prostate cancer aggressiveness [60]. The results from this study agree with the outcome of another pilot study by Wang and coworkers [61].

The glycan analysis of pancreatic ductal adenocarcinoma (PDAC) measured by MALDI–Fourier-transform ion cyclotron resonance (FT–ICR) MSI revealed 8 glycan fragments uniquely expressed in the stroma, and 18 glycan fragments exclusively present in PDAC tumor cells [62]. Sun et al. reported that hyaluronan and chondroitin sulfate overexpression was correlated with worse survival rates, higher concentrations of HexS in stroma were associated with better prognosis, while HexNAcS and HexAHexNAcS abundance predicted worse survival [62].

González de Vega et al. demonstrated that laser ablation inductively coupled plasma (LA–ICP) MSI could be used for the detection of matrix metalloproteinase-11 (MMP-11) to differentiate between the metastatic and nonmetastatic lesions of human breast cancer as a complement to the current approaches. The authors employed prelabelled antibodies with nanoparticles to increase the sensitivity and to facilitate the direct correlation with immunohistochemistry markers [63].

The complexity of the disease renders diverse molecular imbalances, so different studies proceeded to target different molecular classes using a wide range of mass spectrometers. These efforts, such as the ones described here and many others, provide concrete and complementary information to perceive the metabolism of cancer progression.

### 2.3. Mass Spectrometry Imaging in Clinical Diagnosis

From identification of disease-specific markers, prognostic markers, and implementation with machine learning approaches to assist with clinical assignments, mass spectrometry imaging studies provide meaningful contributions to the understanding of the different tumors at the molecular level. However, the methodology is not yet approved for clinical use. We address the current limitations and barriers that need to be overcome before the technology can be accepted into clinical practice in Section 3. Nonetheless, in this section, we discuss recent efforts that showcase the great promise of the technology as a potential tool in routine laboratories.

As previously discussed, the amount of tissue used for diagnosis is limited and often not enough to run the immunohistochemical panels and additional molecular analyses required for a complete diagnostic work up. Especially in biopsy samples, the amount is very limited, and the tumor content present is also sometimes scarce. For this reason, multiplexed approaches where different analytes can be detected from the same sample section are highly attractive. MSI measurements can detect metabolites, peptides/proteins, glycans, and lipids; however, most of the studies consider only one class of analytes. Efforts on maximizing the amount of information obtained from one slide yielded robust protocols that facilitate the measurement of several analytical groups using the same slide [64,65,66,67]. Clift et al. developed a multienzyme workflow for the measurement of extracellular matrix constitution of a single section of FFPE tissue [68]. In this study, sequential digestions with chondroitinase ABC, PNGaseF, elastase, and collagenase Type III were performed. Following each digestion, a matrix was applied, and the sample was measured by MALDI–FT–ICR–MSI [68]. Furthermore, as it is a nondestructive methodology, it is still possible to use the very same section for pathology analysis via histology. The importance of devising such protocols also exceeds the mere fact of saving tissue in routine diagnosis; it opens an unprecedented opportunity to easily colocalize different types of analytes in the very same tissue section, and better understand the biology supporting the molecular changes.

Metal conjugated antibodies have been employed to study the spatial distribution of proteins in the tissue with high spatial accuracy and sensitivity, which is ideal for the quantification of proteins, especially when only residual expression is detected, and therefore to predict the response of a patient to chemical treatment. Along these lines, Bishop et al. resorted to LA–ICP–MSI to simultaneously quantify and localize dystrophin in muscle sections [69]. Duchenne muscular dystrophy is characterized by the absence or decreased expression of dystrophin; to evaluate therapy efficacy, it is necessary to quantify and locate dystrophin in skeletal muscle, but the current methods lack reproducibility and sensitivity. MSI outperformed current techniques with increased sensitivity and using less amount of a sample, which reduces the need for invasive surgical biopsies [69].

The high sensitivity of MSI approaches was also explored for single-cell analysis [70,71,72,73]. While most of the ongoing single-cell studies could provide more information about disease progression, intracellular mechanisms, and novel treatment targets, we can foresee that this level of detail could also be useful in the clinic, especially for early on-set diagnosis, treatment choice, and disease prognosis, for example, to further assess the response of immune cells to a specific treatment, or in the diagnosis of small sections or sections with a low number of tumor cells present, e.g., in small precursor lesions. Using single-cell proteomic characterization, Brunner and coworkers showed that quantifying cellular heterogeneity following targeted perturbation enables the direct analyses of drug responses in single-cell hierarchies on the proteomic level [71]. The study also highlighted the stability of the proteome when compared to single-cell RNA [71]. Further single-cell characterization utilizing MSI was recently reviewed elsewhere [72,74,75].

Another very exciting prospect of this technology was covered in a report by Neuman et al., where they describe the integration of MSI with orthogonal approaches to maximize the information of each experiment [76]. The combination of MSI with microscopy, spectroscopy, transcriptomics, and electrochemistry adds a new layer of information and an exponentially better understanding of the sample. Likewise, achievements in the integration of MSI with spatially targeted tandem MS, the combination of different ionization methods, microextraction, and ion mobility separation are achieving a high level of resolution and opening new possibilities for the technology [76]. While some of the aforementioned multimodal MSI techniques are more useful from a research perspective, others, such as the integration with transcriptomics and microscopy, can revolutionize the way in which clinical diagnosis is conducted.

### 2.4. Mass Spectrometry Imaging for Personalized Medicine

Personalized medicine for tumor assessment has understanding the tumor biology of one person as the main objective. As individual and unique as our DNA, disease expression can also be different from patient to patient, and that is also translated to the response to the treatment. Personalized medicine is devised to maximize the efficiency of disease management considering individual variabilities such as genetics, protein expression, and lifestyle. By utilizing the most state-of-the-art approaches within a relatively short time, accurate diagnosis, prognosis, and therapeutic options are provided to the patient [77,78].

As addressed in Section 2.1, mass spectrometry imaging was utilized to elucidate questions relating to tumor diagnosis and stratification, diagnostic prediction, intratumor heterogeneity characterization, biomarker discovery, and intraoperative consultation. Since the technique allows for such a diverse yet complete overview of tissue composition, MSI has been capturing attention as a convenient approach for personalized medicine.

As the technology is not fully ready to do that just yet, approaches that allow for the integration of current histopathology evaluation and MSI in a single slide can help in moving it toward that direction. The so-called immunohistochemical MSI resorts to photocleavable linkers connected to antibodies that facilitate fluorescent immunohistochemistry (IHC) analysis before performing targeted MSI [79]. An untargeted measurement of the sample is also possible, and it should be performed before adding the antibodies to the sample [79]. Highly multiplexed IHC reduces the number of tissue sections required for a diagnosis. When coupling it to MSI, it is possible to retrieve molecular information directly from the tissue and associate it with high-precision IHC.

## 3. Future Perspectives

There is still much traction to the implementation of more recent technologies in clinical diagnosis, especially when there is integration with machine learning. That resistance is often the result of a lack of understanding on how the said technology works. Machine learning and artificial intelligence (AI) have generally been regarded as competition for human labor and perhaps knowledge for several decades. However, as applications exponentially grow, and their benefits are more palpable, the general acceptance of ML as a reliable aid is also growing. Additionally, the declining number of practicing board certified pathologists in the U.S., Canada, and Europe stands in sharp contrast to a growing demand due to general population growth, an increase in disease incidence, and more complex diagnostics [80,81,82]. Nonetheless, there are still several points that need to be addressed before MSI as a diagnostic technology is able to move forward in the clinic: systematic protocols need to be established and approved for sample preparation and data analysis; more complete and reliable classifiers; user-friendly software; comprehensible outcome of the analysis; and subsequent approval by the competent authorities, such as the European Medicines Agency (EMA) or the Food and Drug Administration (FDA). Additionally, the scientific and medical communities need to work together and combine efforts to vastly increase sample variability that would render this technology applicable across different institutions. As most of the developed classification models are based on samples that are collected and prepared at one institution, an a priori technical bias is introduced. Despite efforts in establishing intercenter studies [80,81,82,83,84,85,86,87,88], it is still necessary to collect samples prepared and measured at different institutions for testing models that were trained independently and achieved similar results.

The majority of developed models have a predefined number of classes, and the outcome of the tissue classification is also one of those classes, as the classification model can only decide between the classes on which the model was trained. For instance, if a sample of gastric cancer would accidentally be classified by a model trained on breast cancer, it would not be able to let the user know that it is not breast cancer. In fact, the outcome infers its breast cancer properties depending on the purpose of the model, even if the sample does not match any of the classes. Hence, it would be important to instate models that can generate another class as an outcome, for example, an “I do not know” or “Are you sure you have selected the right model?” class. The same could also be applied to rarer conditions, or conditions that may have a particular molecular composition. These cases could then be analyzed by a pathologist, who in turn can give the information to further adapt the model. Consequently, this helps in mitigating what we foresee as being another hurdle—the continuous effort to maintain and adapt machine-learning models to the ever-growing number of identified neoplasias, patient prognosis, clinical data, and treatment outcome. The updated datasets used for model training also require strict quality control, especially when the technology is approved for medical diagnosis, to assure the updated model is in line with the applications approved by the competent authorities.

The protocol of a clinical trial plays a key role in the interpretation, management, inspection, and approval to the dissemination of the results. That also should be applied to protocols involving AI to support the assessment of the scientific, ethical, and security issues. To standardize the reporting of AI in clinical trials, SPIRIT-AI has been developed, which provides guidelines for a detailed description of the scope of the trial and the setting in which the trial is evaluated [89,90].

Precision medicine requires, as we demonstrated, the generation of complex data, which also means generation of large datasets, demanding a safe storage solution, computational capabilities for data processing, and reliable data transfer, which is financially very expensive. Nonetheless, the heavy investment should be mitigated over time with a huge benefit for the patient. Associated with data storage limitations are also data protection and ethical guidelines that need to be discussed by the different governmental authorities [91]. In Switzerland, the government took the initiative of creating the Swiss Personalized Health Network (SPHN), which aims to standardize the healthcare information system, and data types between collaborating hospitals and research institutions, to facilitate nationwide health data exchange for health research [92]. The SPHN collaborates with BioMedIT which provides researchers, working in national institutions, with access to a secure and protected computing environment, facilitating the analysis of sensitive data [92]. Further involvement from the government to standardize data collections, and incentivize collaboration between hospitals and research institutions, while keeping data privacy a priority might be the necessary solution. Alternatively, privately owned AI-based imaging approaches are also being approved to provide some patients with a more complete and personalized diagnosis [93]. Arterys received the first-ever US FDA clearance for leveraging cloud computing and deep learning in a clinical setting. They provide a computing cloud with AI-based solutions for analysis of imaging data [94]. Most of the opportunities currently available for AI-based diagnostic approaches are focused on imaging data. While images are an essential part of the diagnosis, it still does not uncover the whole picture, and for that reason, more opportunities could arise for computing of genetic, mass spectrometry imaging, and imaging data using the same interface.

In mass spectrometry, and especially mass spectrometry imaging, a vendor-independent file format, imzML, has been implemented in the scope of the European Union-funded project COMPUTIS [95,96,97,98]. This format facilitates the exchange of MSI data. There are also a few data repositories for MSI: for proteomic data, the Proteomics identification database (PRIDE) has been developed in cooperation with ProteomeXchange (PX) consortium [99,100]; and METASPACE was developed as a metabolite data repository with high-throughput annotation [101,102] and it also contains glycan and lipid databases. The curation of these databases is essential as it allows for faster analysis and an agile comparison between datasets. Moreover, the public availability of the data collected at different institutions has the potential for the creation of more robust classifiers.

While MSI associated with ML tools can provide a large amount of information, it still does not cover all biological interactions in each individual. To have a more complete image, MSI data could be integrated with other approaches with complementary information, such as genomics and transcriptomics (Figure 3) [103,104]. However, data integration is not trivial and requires data repositories carefully curated. Several integration tools are continuously being developed and improved (a summary of the available tools was conducted by Subramanian et al. [103]. Graw et al. also highlighted some of the tools available for integration of biological systems and disease [104]), but deciding which the best tool is for the type of data collected is yet another dilemma. Due to the lack of golden standards and standardized protocols, it is difficult to evaluate the quality of the collected data. In addition, the user needs to consider the compatibility between data types and decide if the process of data normalization and filtering is adequate [104]. This last step requires some knowledge of bioinformatics and data integration tools. Success in creating user-friendly software that facilitates data integration has been limited, as the majority of the available tools are only accessible to researchers with a significant bioinformatics background. In a methodical study to evaluate accuracy, robustness, and computational efficiency of different integration protocols, Duan and coworkers employed 10 integration methods for cancer datasets [105]. The authors also evaluated the influence of combining different omics data types and concluded that integrating multiple datasets does not always result in a better performance for tumor subtyping [105]. The results obtained in this study demonstrate that multiomic data integration shows great promise, and that a better understanding of the types of data to be integrated and the methods employed should be carefully considered, highlighting the demand for further studies on omic data integration.

## 4. Conclusions

We are experiencing the beginning of an exciting revolution in medicine, where novel methodologies are being applied in favor of a precise, individual, and fast diagnosis. Mass spectrometry imaging facilitates the direct molecular characterization of tissue without the need for external targets. MSI protocols have also been expanded to include the measurement of several analytes, still using the same sample tissue, vastly increasing the amount of information obtained from one single slide. Complementarily, integration between different omics approaches, such as genomics, transcriptomics, miRNAomics, proteomics, and metabolomics, provides complementary information toward a more complete understanding of the disease. By carrying out the tissue typing of patient tissue, we envision that MSI would provide a reliable molecular overview of the tissue, find specific therapeutic markers, and assess the response utilizing a very small amount of sample.

The developments considered herein move the technology to a pivotal position for personalized medicine. There are still several challenges to be overcome before the technology can be implemented in medical institutions, but with the great combined effort from the scientific and medical communities, within a few years, the technology could be used to assist in patient diagnosis and prognosis. New endeavors should focus on establishing nationwide guidelines for standardized data collection and facilitate the data transfer between researchers without compromising the patient’s privacy. Furthermore, the integration of proteomic analysis, namely, MSI, with other databases that are being built to advance digital pathology enables a broader understanding of the disease progression, allows for better markers to be developed, and consequently better therapy, while expanding the available machine-learning tools to keep improving the personalized care for every individual patient.

## Figures and Tables

**Figure 1 life-12-01037-f001:**
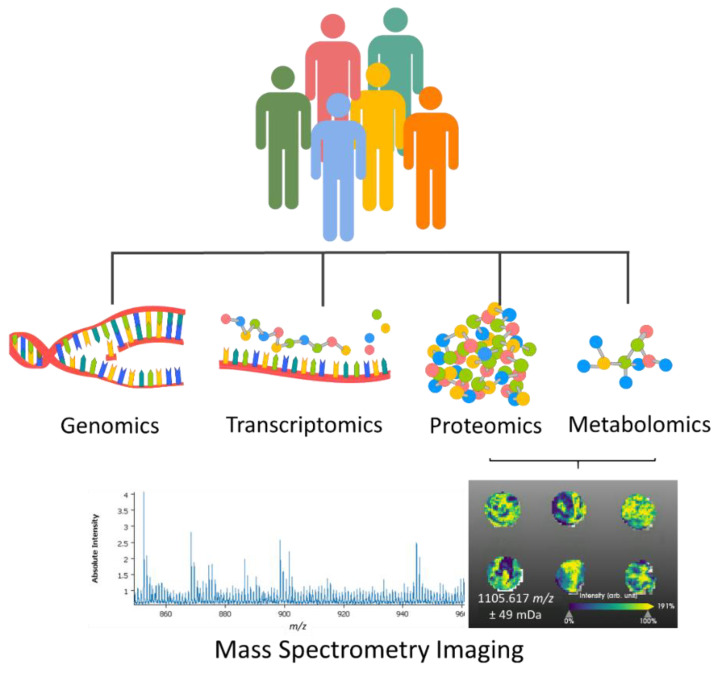
In order to better predict how a patient would respond to a treatment, it is important to precisely evaluate the patient information, which goes beyond the histomorphological characteristics of the tissue. Pathological evaluation with omics approaches, which entail genomics, transcriptomics, proteomics, and metabolomics, gives a more complete understanding of the disease, helps in predicting the response to treatments, and overall improves the patient recovery journey. For the molecular evaluation of tumors, mass spectrometry imaging has been proving its capabilities, ease of integration with histology, digital pathology, and fast measurement times.

**Figure 2 life-12-01037-f002:**
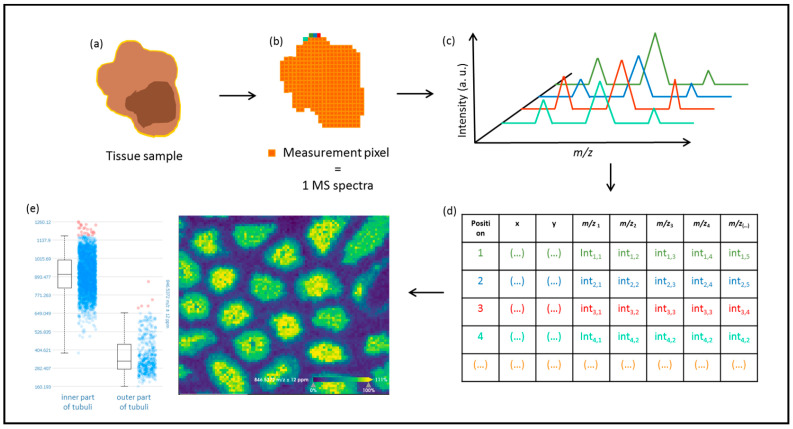
Mass spectrometry imaging from data acquisition to data analysis. (**a**) The selected tissue, after undergoing a sample preparation procedure, is measured directly by mass spectrometry where the defined resolution results in a measurement area or pixel. (**b**) Per each measurement pixel, (**c**) one mass spectrum is recorded. Data are then preprocessed and can be translated into a matrix table (**d**) to facilitate data analysis. The distribution of the intensity of different *m*/*z* values can be easily visualized over the measurement regions using dedicated programs or applications (in this case, SCiLS Lab (Bruker Daltonics, Bremen, Germany) was employed), (**e**) and specific *m*/*z* features can be extracted and further characterized.

**Figure 3 life-12-01037-f003:**
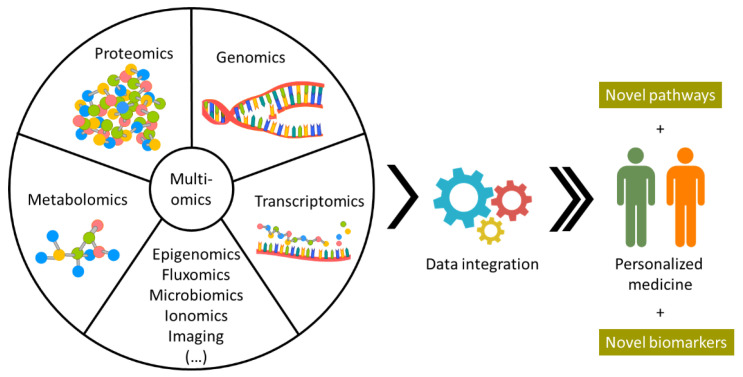
Omic data integration is perceived as the key to unlocking an unprecedent level of knowledge on how different tumor entities develop, better evaluating each individual, and providing a more concrete diagnosis and personalized treatment.

## Data Availability

Not applicable.

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
