# Peer review of "Mass Spectrometry Imaging Spatial Tissue Analysis toward Personalized Medicine"

_life, 2022, doi:10.3390/life12071037_

Round 1

Reviewer 1 Report

The authors have put together an excellent discussion of the current use of mass spectrometry imaging for clinical sample analysis and personalized medicine.  The article will be quite useful for readers wanting to learn more about clinical MSI.  The manuscript is suitable for publication in Life with a few minor revisions as outlined below:

1)      Page 3, lines 96-97 – The sentence beginning with “ The outcome of an MSI run” is awkwardly worded, consider rephrasing.

2)      Page 5, second paragraph – The beginning of the paragraph talks about the use of MSI-related technologies in the operating room, but concludes with an example of a DESI study.  A MasSpec Pen or iKnife example seems more appropriate here.

3)      Reference 71 – The DOI is given twice and the journal name is missing.

Author Response

The authors would like to acknowledge the reviewer’s remarks, which we have taken into account and consider that have contributed to improving the readability of the manuscript.

Herein is a point-by-point response from the authors addressing the reviewer's concerns.

  1. Thank you for pointing out the lack of readability of the sentence. We have therefore adapted it.
  2. We acknowledge the reviewer’s comment. The work we mentioned from Livia Eberlin and co-workers on PDAC tumor margin evaluation seems to be a precedent study for a more recent work using the MasSpec Pen. In light of the reviewer’s comment, we have now decided to highlight the more recent efforts in that paragraph instead of the earlier work.
  3. Thank you very much for noticing the error. We have corrected the reference.

Reviewer 2 Report

The review article entitled “Mass Spectrometry Imaging Spatial Tissue Analysis for Personalized Medicine” by Gonçalves, Bollwein and Schwamborn describes primarily the usage of mass spectrometry imaging (MSI) in the context of cancer diagnostics and characterization. While a review article on this subject would be beneficial to publish, the submitted manuscript requires some changes to be informative enough for publication.

General comments:

Referencing throughout the manuscript is quite sparse, with multiple statements left without citations. The authors should make another pass through the text and ensure all assertions are supported by references.

The figures require additional work. Figures 1 and 2 contain panels clearly reproduced from primary MSI publications. These publications should be properly referenced. Additionally, the text contained on these panels is too small to be legible. It would also be beneficial to consider additional figures, particularly demonstrating the agreement between immunohistochemistry and MSI which is alluded to multiple times throughout the text but not visually supported.

Specific comments:

Section 2, “Mass Spectrometry Imaging” – the first paragraph (lines 60 to 73) has nothing to do with MSI and should be moved up to the introduction.

Figure 2 – the reference to ‘laser resolution’ is only specific to MALDI-MSI and related techniques. Please revise.

A crucial aspect of intraoperative tissue analysis is speed – while techniques specifically developed for that purpose (e.g. the iKnife) allow real-time analysis, the vast majority of traditional MSI approaches are quite time-consuming (more so at high spatial resolution) and thus unsuitable for this task. A clearer delineation between these two types of MSI throughout the text would help.

Perhaps a more general comment related to the above - referring to just MSI rather than specific techniques throughout the text is not sufficiently informative (e.g. line 218); every study description should include the name of the specific MSI technique employed.

Related to the above point - the number of available MSI techniques has tremendously increased in recent years, particularly among atmospheric-pressure approaches. It would be useful to compare the most commonly used MSI approaches, perhaps in a table, listing the acronyms, spatial resolution and analyte class compatibility, along with references to primary publications. Also, a list of acronyms at the end of the article may be useful to consider.

Paragraph starting with line 89 – the description of the fundamentals of mass spectrometry is inaccurate. Additionally, line 90, reference number 18 is an inappropriate self-citation for the description of a basic mass spectrometer. The authors should find a more fundamental text to cite. The “separation module” is almost universally referred to as the mass analyzer, to contrast with external “separation modules” such as chromatography or various forms of ion mobility separation. Regardless of the type of mass analyzer, the analytes are not separated by mass, but by their mass-to-charge ratio from which a mass value can then be derived (rather than the other way round as implied). This entire section needs careful revision.

Line 351 – considering that this line makes it clear that MSI is not currently being used for personalized medicine and isn’t predicted to become a staple clinical tool for at least a few more years, the title of the manuscript may be inappropriately optimistic. Consider revising to “Mass Spectrometry Imaging Spatial Tissue Analysis: Towards Personalized Medicine” or something along those lines.

Author Response

Firstly, the authors would like to acknowledge the remarks from the reviewer, which we have carefully considered. Considering the references, we appreciate the comment, we have revised it and added references where we considered necessary.

As for Fig 1 and Fig 2, the examples added are from original measurements that have not been published so far. Moreover, all the graphics have been produced by the first author. We understand the reviewer’s concerns relating to the font size, which have been addressed within the realm of possibility to not distract from the overall schematics. Furthermore, we take the chance to remind the reviewer that the publication will be mainly available online and that the images can be better seen in the digital version with the possibility of zooming in.

Herein is a point-by-point response from the authors addressing the reviewer’s specific comments.

Specific comments:

Section 2, “Mass Spectrometry Imaging” – the first paragraph (lines 60 to 73) has nothing to do with MSI and should be moved up to the introduction.

R: We acknowledge the reviewer’s criticism, but performing this change will in the author’s opinion worsen the readability of the publication. The paragraph mentioned starts by underlining one of the major limitations of MSI as an invasive approach, hence we consider it to be very well fitted in the context of MSI.

Figure 2 – the reference to ‘laser resolution’ is only specific to MALDI-MSI and related techniques. Please revise.

R: Thank you very much for pointing it out. We did indeed aim at including all MSI techniques and not only laser-assisted approaches. It has been corrected.

A crucial aspect of intraoperative tissue analysis is speed – while techniques specifically developed for that purpose (e.g. the iKnife) allow real-time analysis, the vast majority of traditional MSI approaches are quite time-consuming (more so at high spatial resolution) and thus unsuitable for this task. A clearer delineation between these two types of MSI throughout the text would help.

R: We do see validity in the reviewer’s comment and we do understand that for the specific application the time often comes at the cost of resolution (understandably). We have adapted the paragraph to emphasize the speed of the process. However, these approaches rely on robust classifiers that took a long time in sample measurement and analysis in order to be built, similar to other approaches and we also do not want to neglect that incredible effort.

Perhaps a more general comment related to the above - referring to just MSI rather than specific techniques throughout the text is not sufficiently informative (e.g. line 218); every study description should include the name of the specific MSI technique employed.

R: We have added the specific technique employed to the paragraphs where it was missing. However, the mentioned paragraph is the last paragraph of the section, referring to all the techniques mentioned in the section. We purposely included diverse mass spectrometry approaches to show that MSI as a field has been making major improvements.

Related to the above point - the number of available MSI techniques has tremendously increased in recent years, particularly among atmospheric-pressure approaches. It would be useful to compare the most commonly used MSI approaches, perhaps in a table, listing the acronyms, spatial resolution and analyte class compatibility, along with references to primary publications. Also, a list of acronyms at the end of the article may be useful to consider.

R: We thank the reviewer for this valuable suggestion. However, the scope of the paper is to highlight applications and not the different MSI techniques in detail. This has been done previously, in particular with regard to ambient ionization techniques. Therefore, we have added a short paragraph to inform the readers about the ever-growing techniques and have referred them to a publication by Feider et al.  (https://doi.org/10.1021/acs.analchem.9b00807).

Paragraph starting with line 89 – the description of the fundamentals of mass spectrometry is inaccurate. Additionally, line 90, reference number 18 is an inappropriate self-citation for the description of a basic mass spectrometer. The authors should find a more fundamental text to cite. The “separation module” is almost universally referred to as the mass analyzer, to contrast with external “separation modules” such as chromatography or various forms of ion mobility separation. Regardless of the type of mass analyzer, the analytes are not separated by mass, but by their mass-to-charge ratio from which a mass value can then be derived (rather than the other way round as implied). This entire section needs careful revision.

R: We thank the reviewer to bring the concern to our attention. We do see how it could be misinterpreted. We have adapted accordingly.

Line 351 – considering that this line makes it clear that MSI is not currently being used for personalized medicine and isn’t predicted to become a staple clinical tool for at least a few more years, the title of the manuscript may be inappropriately optimistic. Consider revising to “Mass Spectrometry Imaging Spatial Tissue Analysis: Towards Personalized Medicine” or something

R: We agree with the reviewer’s suggestion; hence it has been adapted.